# Fine Classification Method for Massive Microseismic Signals Based on Short-Time Fourier Transform and Deep Learning

**Chunchi Ma** [1], **Xuefeng Ran** [1,*], **Weihao Xu** [1], **Wenjin Yan** [1], **Tianbin Li** [1], **Kunkun Dai** [1], **Jiangjun Wan** [2], **Yu Lin** [1] **and Ke Tong** [1]

1    State Key Laboratory of Geohazard Prevention and Geoenvironment Protection, Chengdu University of Technology, Chengdu 610059, China
2    School of Architecture and Urban-Rural Planning, Sichuan Agricultural University, Chengdu 611830, China
*    Correspondence: jamesbond@stu.cdut.edu.cn

**Abstract:** Numerous microseismic signals are produced by rock mass fracture during earthquakes, geological disasters, or underground excavations. Moreover, a large amount of noise signals are captured during microseismic signal monitoring. Specifically, some noise signals closely resemble microseismic signals, which severely impedes the rapid and accurate detection of the latter and the assessment of geological hazards. Therefore, we propose a precise model for identifying and classifying microseismic signals based on deep learning technology and short-time Fourier transform (STFT) technology. First, the STFT time–frequency analysis reveals the unique characteristics of noise, microseismic, and blasting signals, thereby allowing noise signals that are very similar to microseismic signals in the time domain to be finely distinguished. Second, the introduced attention mechanism focuses the classification on essential signal features. Finally, because tens of thousands of actual monitoring data points are considered, the deep neural network for microseismic classification is trained and tested under complex geological engineering conditions. The results demonstrate that the neural network model has good time–frequency feature extraction ability, and the well-trained model can satisfactorily complete daily classifications. Moreover, the model performs well when classifying similar noise and low-SNR microseismic signals. We believe that this type of signal-processing method, which considers multiple perspectives, can be extended to data processing in many other data-driven fields.

**Keywords:** microseismic signal; classification; time–frequency analysis; attention mechanism; deep learning

## 1. Introduction

As a novel technology for evaluating rock mass stability, microseismic monitoring has been widely utilized in underground mining [1–3], underground powerhouses [4], tunnel excavation [5–7], and other projects in which rock mass stability is crucial [8,9]. It has grown in popularity due to its good disaster-warning abilities, ensuring smooth construction workflow and saving construction costs. Microseismic monitoring aims to quantitatively evaluate the stability of the rock mass by detecting specific elastic waves generated during rock microfracture [10–12]. However, this technology is susceptible to environmental influences. Moreover, its application environment is generally complex and changeable. Therefore, different types of signals are received by the sensor, including unique noises (they will be introduced in Sections 2.3 and 4.2.2, under similar noise (SN)), which refer to signals that resemble the low-SNR microseismic signals in the waveform. In practice, accurately extracting microfracture signals from the daily mass of signals, even for experienced and professional classifiers, is time-consuming. Notably, ensuring accuracy in the presence of unique noise is difficult. A multiplicity of factors hinder classification and make efficiently identifying microseismic signals and the precise location of the focal point

a global problem [13]. Therefore, a fast and accurate method of classifying microseismic signals is urgently required.

Scholars and experts at home and abroad have intensively sought to improve the accuracy and efficiency of microseismic signal classification. Microseismic signal recognition methods are divided into multiparameter joint and time–frequency analysis methods. Dong et al. [14] established a sample database of mine blasting and microseismic events through statistical analysis and established a statistical model for automatically identifying multicharacteristic indicators. Vallejos [15] proposed a classification algorithm for mine microseismic and blasting signals based on multisource parameters and a logistic neural network. Shang et al. [16] proposed an artificial neural network based on principal component analysis to identify microseismic signals and mine-blasting signals. In recent years, the time–frequency analysis method has been increasingly applied in microseismic signal identification, and derived techniques are continually emerging [17–20]. Lu et al. [21] analyzed the power and frequency spectra of microseismic signals through Fourier transform and proposed a method of identifying microseismic signals in the waveform. Empirical mode decomposition (EMD) is a type of adaptive decomposition algorithm [22]; the signal is decomposed into a finite number of intrinsic mode function (IMF) components and a residual term. This method is theoretically applicable to any signal. Shang et al. [23] proposed a feature extraction and classification method and applied it to mine microseismic and blasting signals based on EMD and singular-value decomposition. They then used a support vector machine for classification and achieved an accuracy of up to 93%. Zhu et al. [24] utilized wavelet packets to decompose signals, obtained fractal box dimensions in specific frequency bands, and determined that different signals had distinct fractal characteristics that could be exploited as the basis of classification. Zhao et al. [25] analyzed the time–frequency characteristics of microseismic signals and blasting signals and studied the distribution characteristics of energy in subfrequency bands by applying the frequency-slicing wavelet transform method, which constituted a novel application to microseismic signal recognition. The rapid development of computer technology has improved all aspects of human life. Big data processing, artificial intelligence, and other hot technologies, including microseismic monitoring data processing, have been introduced to the geological industry for data processing. These methods involve machine learning techniques. Zhang et al. [26] and Zhao et al. [27] proposed a three-classification (microseismic, blasting, and noise) and a multiclassification deep learning neural network model, respectively, for microseismic signals: both achieved an accuracy of more than 90%. Ma et al. [28] proposed a novel classification model based on bimodal neurons in an ANN (artificial neural network), an exemplar of deep learning models identifying signals from multiple perspectives.

Although the above methods have achieved success in identifying microseismic signals, they are not without their shortcomings. For instance, although the multiparameter joint analysis method has high accuracy, it is not suitable for rapid and real-time monitoring and early warning due to its numerous parameters and complex operation. The fast Fourier transform in time–frequency analysis encounters challenges in describing the local signal characteristics. The EMD decomposition exhibits excessive decomposition and faulty components. In addition, the above methods are mainly applicable to distinguishing mine microseismic signals from blasting signals, with only a few applications in tunnel microseismic monitoring. Moreover, the classifiers become confused when unique noise signals that are highly similar to the rock microfracture signals are present in the waveform, which affects the accuracy of event classification and warning efficiency.

To achieve efficient and reliable classification, we utilize the short-time Fourier transform (STFT) to process the primary data on rock microfracture signals, high-similarity noises, blasting signals, mechanical noises, and environmental noises in the tunnel. Furthermore, a deep learning convolutional neural network with an attention mechanism is utilized as the classifier to learn and test tens of thousands of signals after they are processed via the STFT method. In addition, the model without STFT processing is used

as the baseline to evaluate the performance of the key model. The results demonstrate that the model based on time–frequency domain identification is significantly superior to that based on time domain identification in both the recognition and classification of a single signal and the joint recognition of microseismic events from multiple signals. The recognition of processing signals from the time and frequency domain by utilizing artificial intelligence has great application prospects in fields such as seismic exploration, remote sensing, and other areas where efficient signal processing is a requisite.

## 2. Methods and Data Preparation

### 2.1. Short-Time Fourier Transform (STFT)

Fourier transform is a common spectrum analysis method used for signal processing. By applying the Fourier transform to the waveform signal, the distribution of the waveform in the frequency domain can be obtained to extract its frequency characteristics. However, the signal feature in the original time domain will be lost. Short-time Fourier transform (Formula (1)) is a window Fourier transform [29–32]. It can extract characteristics from both the time and frequency domains by splitting the original signal in the time domain into small sections and performing subsequent recollection after extracting the frequency features of each section using STFT. The STFT-processed signal can not only complete the frequency domain feature extraction but also retain the time domain feature to a certain extent.

$$STFT_s(t, \omega) = \int_{-\infty}^{+\infty} S(\tau)h(\tau - t)e^{-i\omega\tau}\mathrm{d}\tau \tag{1}$$

where the $S(\tau)$ function represents the waveform in the time domain and $h(\tau - t)$ represents the window function $\omega = 2\pi f$.

However, this technique is disadvantageous because it cannot wholly and simultaneously retain the frequency domain and time domain information features. When the width of the window function is too large, the frequency resolution will increase, but the time resolution will decrease and vice versa. Therefore, the choice of the window function width is informed by whether the actual requirement emphasizes the time or frequency domain feature. In this study, the length of the window function was set to 256, and the hamming window function was selected.

### 2.2. Analysis of Signal Time-Frequency Characteristics

Numerous and varied complex signals are captured by the sensor, as determined by the construction environment. One or more reliable distinguishing features between microseismic and noise signals are needed to classify them. We studied the characteristics of both waveforms in the time domain and the frequency in the time–frequency domain. Figure 1 shows the waveform and time spectrum of six signals. Notably, a single event contains six waveforms. If three or more waveforms in a single event are identified as microseismic signals, this event is regarded as a microseismic event.

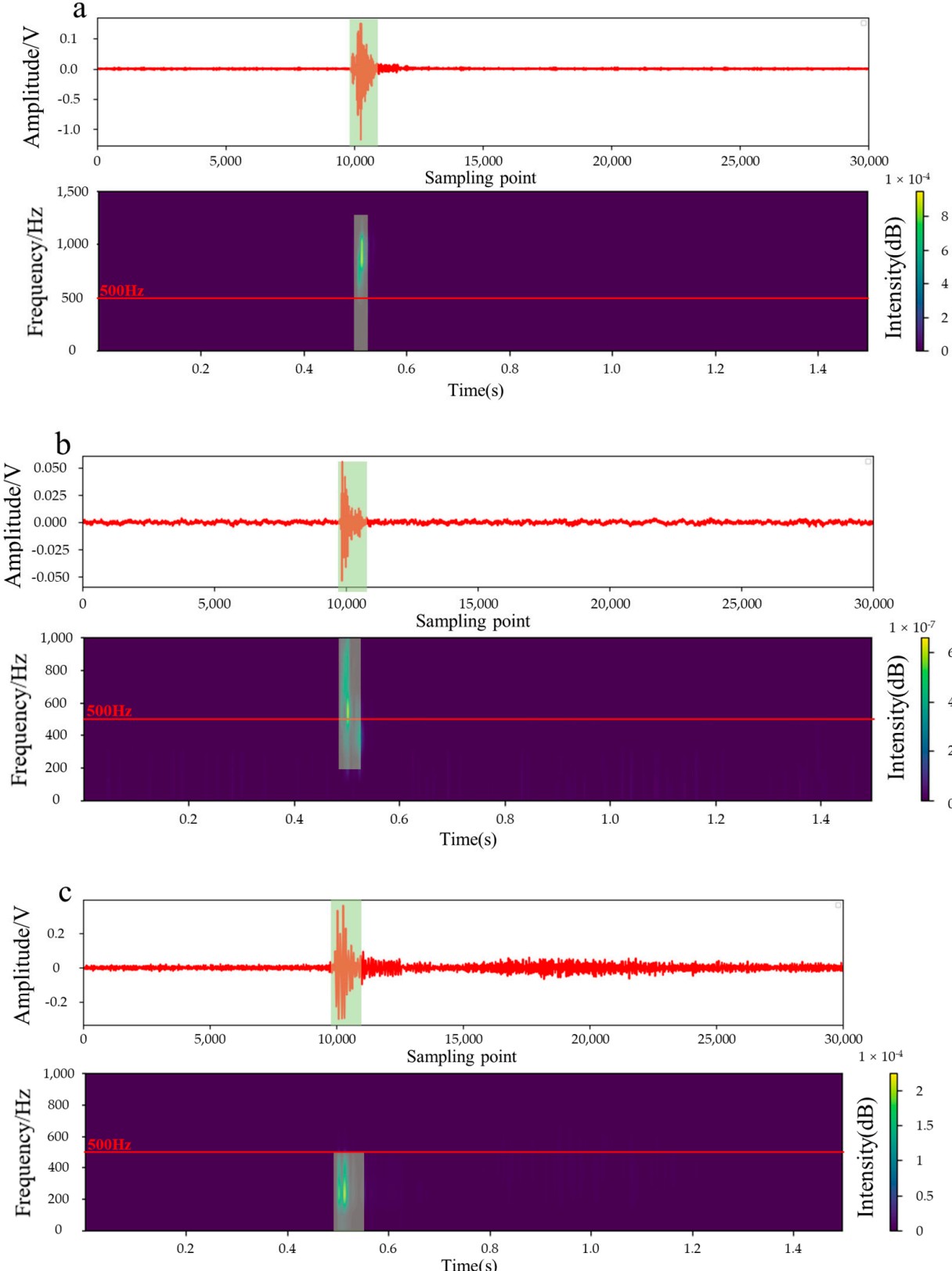

**Figure 1.** *Cont.*

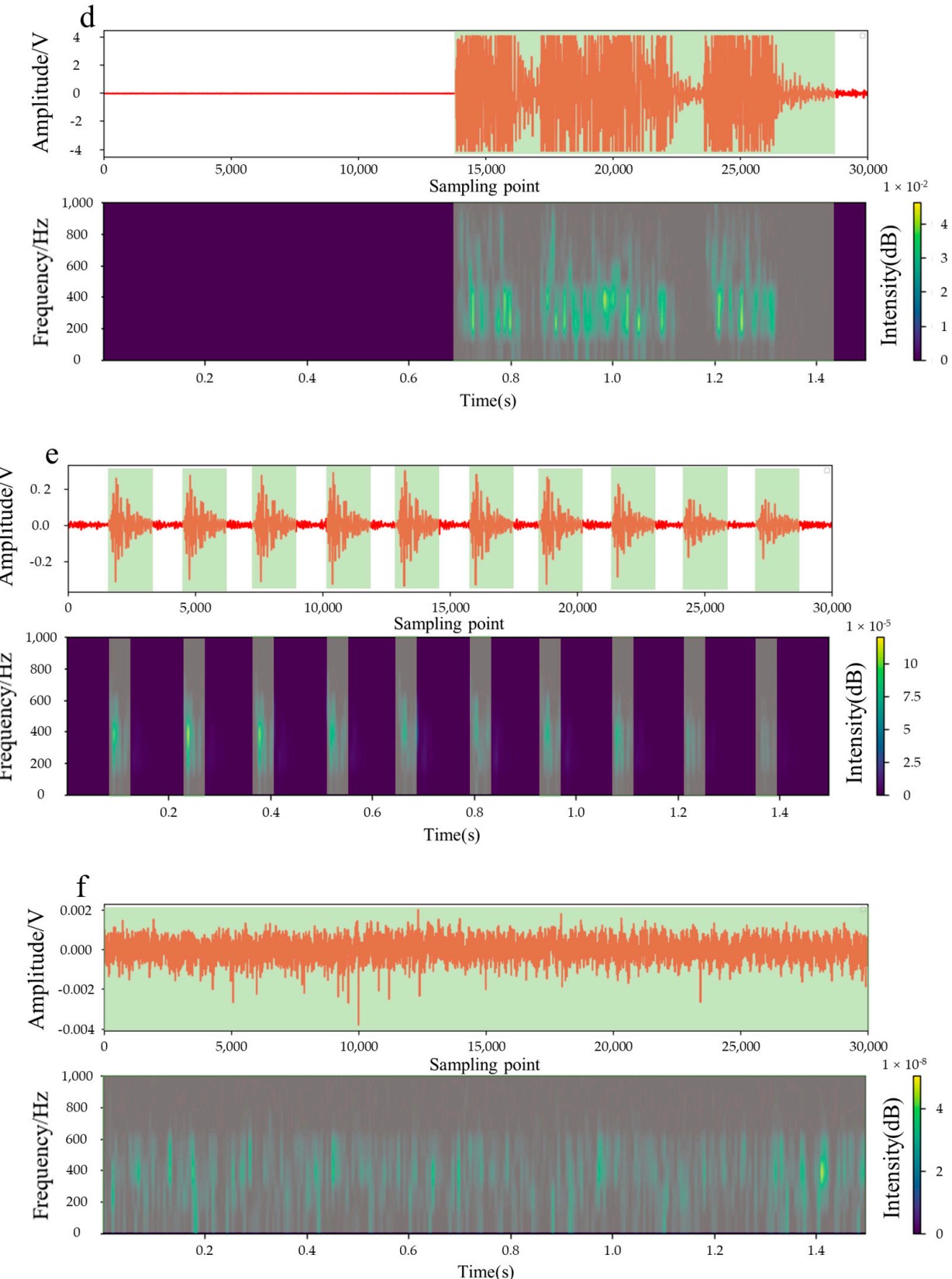

**Figure 1.** Various common signals and their amplitude spectra. The light green area represents the main time domain or frequency domain characteristics of the signal: (**a**,**b**) denote the microfracture signals at high and low amplitudes, respectively. (**c**) is the unique noise signal that is similar to the microfracture signal in (**b**); (**d**–**f**) represent the blast, mechanical, and unknown waveforms, respectively.

Figure 1a,b show microseismic signals with high and low SNR, respectively. They exhibit a brief period of dense oscillation on the waveform then quickly disappear, and the tail wave does not develop. The amplitude ranges from tens to 2000 mV. The microseismic signal from the time spectrum analysis exhibits the most energetic components at frequencies higher than 500 Hz. Figure 1c shows a special type of noise, which is similar to the low-SNR rock microfracture signal in the waveform. Notably, it has a short oscillation duration and maximum amplitude similar to the low-SNR microseismic signal. Therefore, it is termed highly SN. It can easily confuse classifiers and deteriorate classification efficiency when it is only distinguished from time domain images. The time spectrum analysis revealed that this type of noise exhibits the most energetic components at frequencies lower than 500 Hz. Figure 1d shows the shock wave generated by blasting, which is characterized by a long duration and large-amplitude waveform. It also has relatively rich frequency components. Figure 1e shows mechanical noise with prominent characteristics, including regular multiple continuous shocks. Its single shock is also similar to that of the microseismic signal with low SNR. The frequency components are predominantly in the 0–600 Hz range. Figure 1f presents dense and disorderly Gaussian environmental noise, with its duration ranging throughout the signal period. Moreover, the amplitude is generally low and easy to distinguish, but the frequency component is complex.

The aforementioned analysis revealed the following: for the waveform in the time domain, the microfracture signals with low SNR were highly similar to the unique noise signals introduced above, but they differed considerably in frequency in the time-frequency domain. Specifically, the frequency components of the microfracture signal were always higher than 500 Hz, whereas the upper limit for the unique noises was seemingly lower than 500 Hz. In terms of signal duration, the microfracture signal exhibited a single peak, its coda wave following the peak faded sharply, and the duration was extremely concise. Although the unique noise waveforms were similar to the low-SNR microfracture signal waveforms, they often exhibited multiple peaks. Specifically, they differed considerably, both in the time and the time–frequency domains, for blasting, mechanical, and environmental noise signals. Therefore, they are easily identified by the classifier. In summary, distinguishing microseismic signals only from the time domain could remove most noise signals. However, distinguishing microseismic signals with low SNR and special noise is challenging. Therefore, adequate signal classification accuracy can be achieved by utilizing both time and frequency domain features.

*2.3. Attention Mechanism*

Because different signals have different durations, we introduced the attention mechanism to confer adaptive learning abilities upon the network. Accordingly, the PC (Equip with i7-9700K CPU, NVIDIA GeForce 2080 Ti GPU, and 32 GB RAM) can reasonably allocate computer resources to ensure the model can adequately learn the relevant features.

The principle underlying the attention mechanism is borrowed from human beings' tendency toward excessive attention when receiving information. To obtain a complete description, humans focus on the relatively essential and interesting parts of a description. In contrast, the descriptions of relatively unimportant information receive little attention. When an attention mechanism is added to the neural network in machine learning, the computer focuses on learning the relatively important part of the input features during learning. The method involves assigning weights to different feature parts and then allocating computing resources to each according to the proportion of weights. In contrast, the neural network without an attention mechanism allocates the same amount of computing resources to each part of the feature.

The convolutional block attention module [33] attention mechanism was adopted in this study, which can be divided into channel attention (Figure 2) and spatial attention mechanisms (Figure 3).

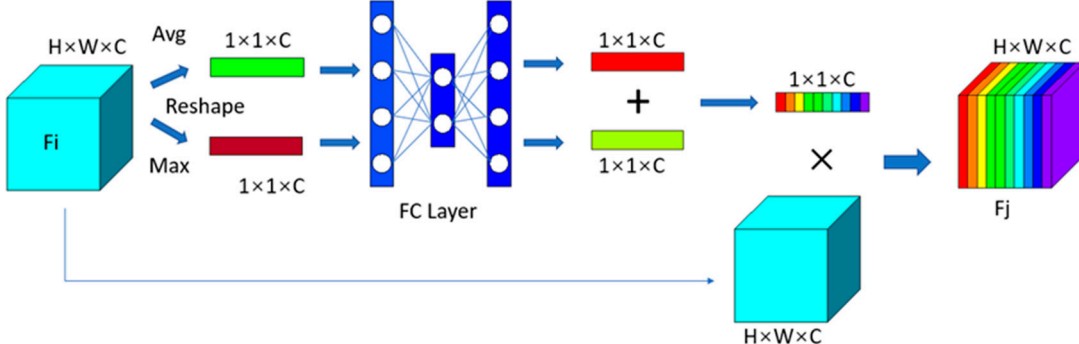

**Figure 2.** Structure of the channel attention module. H, W, and C denote the height, width, and channel, respectively, representing the input size.

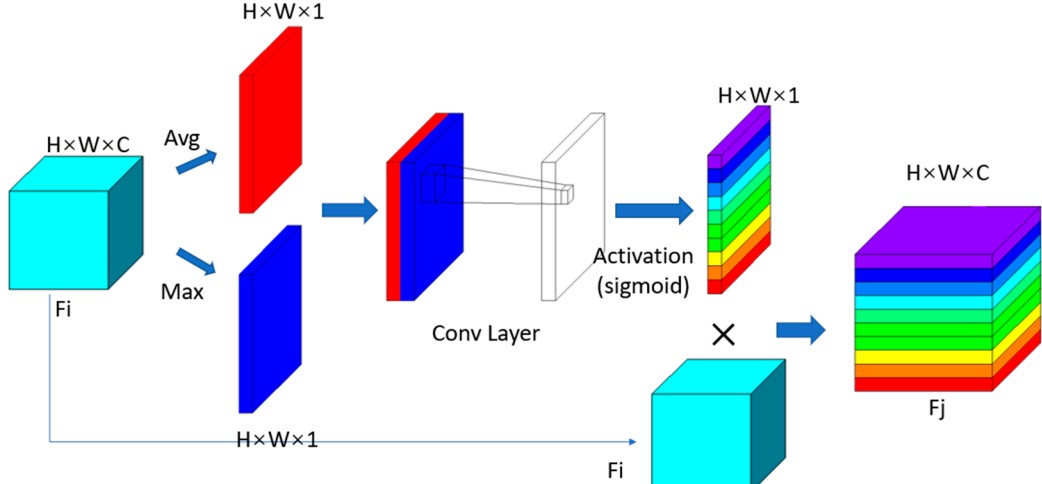

**Figure 3.** Structure of the spatial attention module. H, W, and C are height, width, and channel, respectively, representing the input size.

The channel attention mechanism identifies meaningful features. First, the input features are averaged and maximally pooled. The average pooling reduces the dimension of the input features to calculate the channel attention effectively. The maximum pooling highlights the uniqueness of the input features and refines the attention on the channel. Two feature maps were obtained through the fully connected network after pooling and then superimposed, and the sigmoid activation function was applied to ensure the sum of weights of each part was equal to 1. Finally, after the feature matrix, before the input was multiplied with the obtained weight matrix, the weight of each part of the feature matrix was successfully assigned. That is, the feature was strengthened in the relatively important part and weakened in the relatively unimportant part.

Spatial attention was focused on the most informative part, which complements channel attention. First, average and maximum pooling operations were applied along the channel axis, then the two feature maps were concatenated together to produce a valid feature descriptor. Thereafter, the convolution layer was utilized to generate a spatial attention map that codes locations that need to be emphasized or suppressed. Finally, after the primary input was multiplied by the obtained weight map, the feature with weight distribution was generated.

### 2.4. VGG13 Modified Network

VGG Net, a deep convolutional neural network proposed by the Visual Geometry Group of Oxford University [34], was the runner-up in the 2014 ImageNet Competition. Its network structure is neat and very suitable for hardware acceleration.

We modified the standard VGG13 (10-layer convolution + 3-layer fully connected) network structure to serve as the network model for this training. Figure 4 shows the adapted network structure: the convolutional layer extracted data features through convolution calculation, the convolution kernel was set to 3 × 3, and the sliding step was 1. The pooling layer reduced the amount of feature data, and the maximum pooling was selected. The pooling kernel was set to 2 × 2, and the sliding step was consistent with the pooling kernel. The batch normalization operation was used to distribute the small-batch feature data into the linear region of the activation function, thus improving the discrimination power of the activation function with respect to input data; the rectified linear activation function (ReLU) function was used as the activation function. A dropout operation was used to temporarily discard a certain proportion of neurons randomly in the hidden layer (0.35 in this experiment) during training, which can alleviate the overfitting of the neural network and improve the generalization ability of the model. The optimizer tool (Adam in this study) is used for configuring the training method. It guides the neural network to update the parameters, and the categorical_crossentropy function was selected as the loss function, which is used to evaluate the modular learning effect. The learning rate was set to 0.01. The SoftMax activation function was used to ensure that the output result was in the probability distribution of each category. The optimal model was generated and saved by minimizing the loss value of the test set results during training. The network structure was adapted from VGG13. The total number of convolutional layers was 12, and the channel and spatial attention mechanism modules were connected after every two consecutive convolution layers. Finally, one fully connected layer was used as the output layer after flattening the multidimension vectors. Unlike in the VGG13, a two-consecutive-convolution layer was used to replace two fully connected layers, thus improving the feature extraction ability of the network model and reducing the training parameters. A higher number of network layers improved the feature extraction effect.

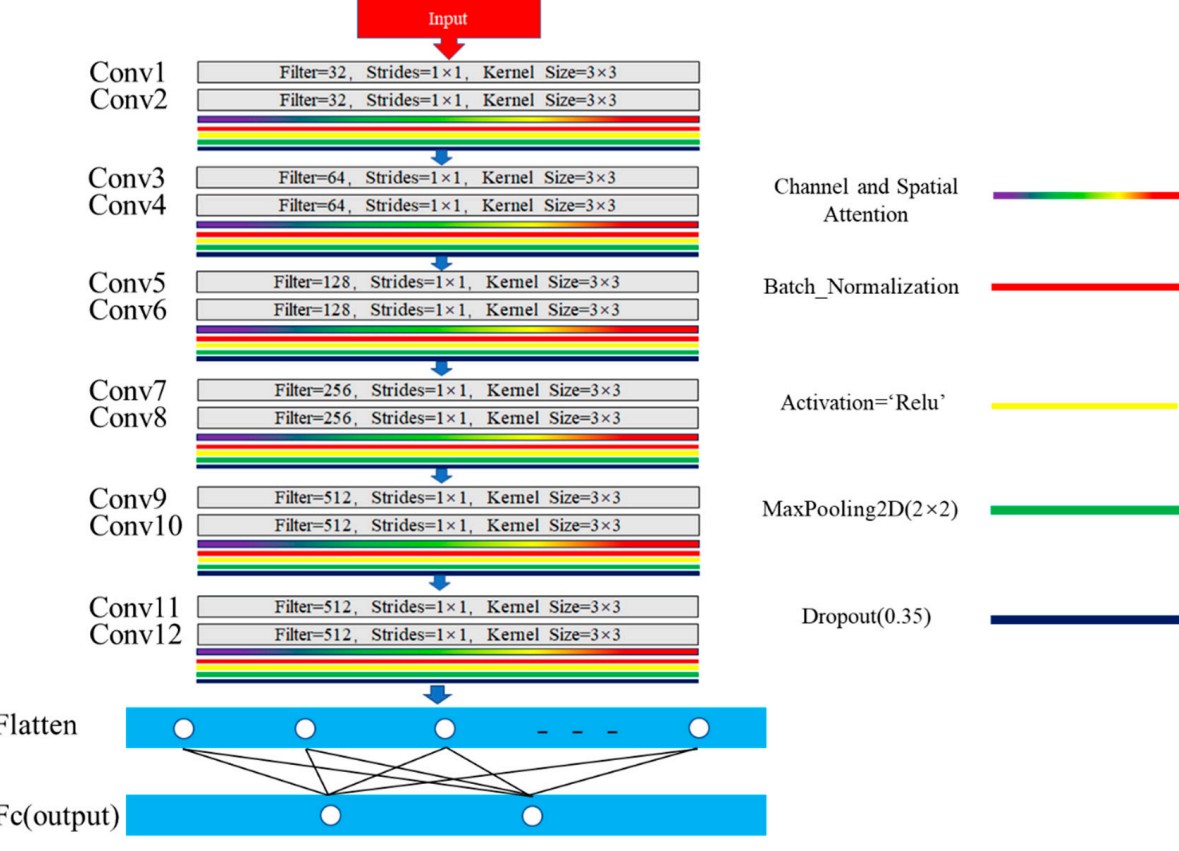

**Figure 4.** Modified network structure based on VGG13.

*2.5. Dataset*

The single signal data type received by the microseismic monitoring equipment is (30,000, 1), a one-dimensional vector composed of 30,000 digits. The single-signal data type after STFT conversion is (129, 236, 2). The data set contains 13,824 time–frequency signal data points after STFT transformation, including 6616 microseismic data points and 7208 noise data points (500 SNs). The data set was randomly divided into training and validation sets: 11,059 data points in the training set contained 5308 microseismic data points and 5751 noise data points (SN should be approximately 400 pieces), and 2765 data points in the validation set contained 1308 microseismic data points and 1457 samples of noise data points (SN corresponds to approximately 100 noise data points).

Table 1 lists the output categories and number of parameters of each layer of the neural network model (time–frequency microseismic classification (TFMC)). Excluding the attention mechanism layer (which does not change the input feature dimension), the total number of network layers was 13, 12 of which were convolution. The latter were then connected to a fully connected output layer after straightening. Notably, the number of training parameters can reach $9.44 \times 10^6$.

**Table 1.** Parameters of the TFMC model.

| Layer (Type) | Output (Shape) | Param |
|:---:|:---:|:---:|
| Input | (None, 129, 236, 2) | 0 |
| Conv1 | (None, 129, 236, 32) | 608 |
| Conv2 | (None, 129, 236, 32) | 9248 |
| Maxp1 | (None, 65, 118, 32) | 0 |
| Conv3 | (None, 65, 118, 64) | 18,496 |
| Conv4 | (None, 65, 118, 64) | 36,928 |
| Maxp2 | (None, 33, 59, 64) | 0 |
| Conv5 | (None, 33, 59, 128) | 73,856 |
| Conv6 | (None, 33, 59, 128) | 147,584 |
| Maxp3 | (None, 17, 30, 128) | 0 |
| Conv7 | (None, 17, 30, 256) | 295,168 |
| Conv8 | (None, 17, 30, 256) | 590,080 |
| Maxp4 | (None, 9, 15, 256) | 0 |
| Conv9 | (None, 9, 15, 512) | 1,180,160 |
| Conv10 | (None, 9, 15, 512) | 2,359,808 |
| Maxp5 | (None, 5, 8, 512) | 0 |
| Conv11 | (None, 5, 8, 512) | 2,359,808 |
| Conv12 | (None, 5, 8, 512) | 2,359,808 |
| Maxp6 | (None, 3, 4, 512) | 0 |
| Flatten | (None, 6144) | 0 |
| FC | (None, 2) | 12,290 |

## 3. Results

*3.1. Evaluation Indicators*

A good model should have good generalization ability. Moreover, a unified standard is required to evaluate the generalization ability of different classification models. Different tasks have different performance measures; recall, precision, and F1 score are generally used to evaluate model generalization for binary classification tasks. Herein, a comprehensive performance comparison of the model was conducted via the receiver operating characteristic curve (ROC) and area under the ROC curve (AUC) based on the true positive rate (TPR) and the false positive rate (FPR).

In a dichotomous task, the results predicted by the learner can be divided into four categories: true positive (TP), false positive (FP), true negative (TN), and false negative (FN). Microseismic and noise signals are positive and negative examples in signal classification, respectively, corresponding to TP and TN, which the learner would identify correctly. Furthermore, FP and FN represent microseismic and noise signals that the learner would

wrongly identify. Recall, precision, and F1_score are defined in Table 2. (Micro F1_Score is the weighted harmonic average of precision and recall on a single class, and Macro F1_Score is the arithmetic mean value calculated for multiple Micro F1_Scores when multiple classes are present. The "n" in the Macro F1_Score definition formula denotes the number of classes.)

**Table 2.** Evaluation indicators.

| Name of Indicator | Definition Formula |
|---|---|
| Precision | $P = \frac{TP}{TP+FP}$ |
| Recall | $R = \frac{TP}{TP+FN}$ |
| Micro F1_Score | $Micro\ F1\_score = \frac{2 \times P \times R}{P+R}$ |
| Macro F1_Score | $Macro\ F1\_score = \frac{1}{n} \sum_{i=1}^{n} Mirco\ F1\_score_i$ |
| TPR | $T = R = \frac{TP}{TP+FN}$ |
| FPR | $F = \frac{FP}{TN+FP}$ |

*3.2. Training Results*

Because the time–frequency model network structure was adapted from VGG, the VGG13 and VGG16 networks were selected for comparison. Because the two-layer full connection was removed from the time–frequency model network structure based on VGG13 and replaced with two-layer convolution, the trainable parameters were the least, the loss value of the validation set was the least, and accuracy was 0.993 (Table 3).

**Table 3.** Comparison between different networks. Val_Loss and Val_Acc represent the loss and accuracy for the validation dataset, respectively.

| Model | Parameters ($\times 10^6$) | Val_Loss | Val_Acc |
|---|---|---|---|
| VGG13 | 9.97 | 0.034 | 0.991 |
| VGG16 | 13.04 | 0.032 | 0.993 |
| TFMC | 9.44 | 0.022 | 0.993 |

Increasing the training epoch gradually improved the accuracy in the training set (above 98%), and the loss value continuously decreased (below 0.01) before stabilizing. However, the accuracy and loss values of the validation set fluctuated wildly (Figure 5). We only retained the model with the best fitting of the learner on the test set: the total number of iterations was 300. The model was optimized after the 137th iteration, the accuracy of the test set was 99.3%, and the loss value decreased to 0.02. The set results of the classification validation are listed in Table 4.

**Table 4.** Confusion matrix of the validation dataset. MS and NS represent the microfracture signal and noise signal, respectively.

| Prediction<br>Classes | MS | NS |
|---|---|---|
| MS | 1266 | 42 |
| NS | 32 | 1425 |

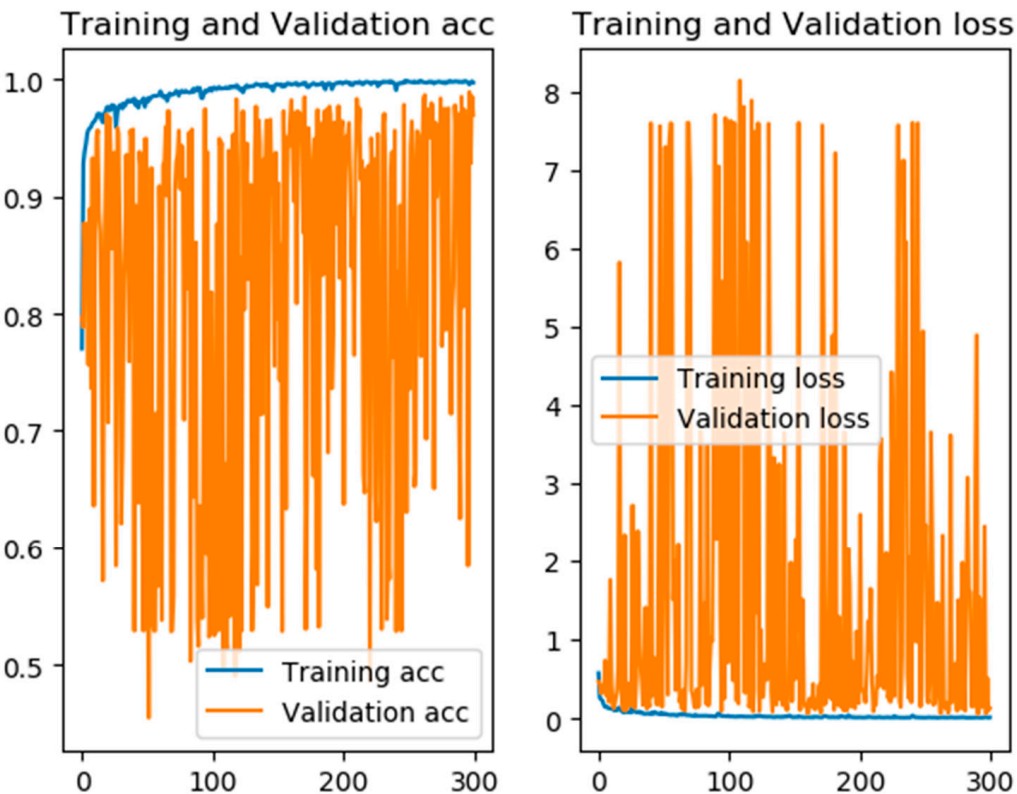

**Figure 5.** Model training process and results. The left figure denotes the accuracy changes with epochs, whereas the right one denotes loss changes with the epochs. The bule and orange lines represent the accuracy and loss for the training and validation datasets, respectively.

Among the 1308 microseismic signals and 1457 noise signals tested, the learner correctly classified 1266 microseismic signals and 1425 noise signals with a precision, recall, and F1-score of 97.5%, 96.8%, and 97.1%, respectively.

### 3.3. Model Comparison

To better evaluate the generalization of the model, we picked an extra dataset consisting of 500 microseismic signals, 500 blast noise signals, and 500 Gaussian noise signals. The test data never appeared in the training or validation sets, which were not utilized during training. However, they were utilized for evaluating the generalization of the model. The time domain model (TMC), a microseismic signal classification model, was trained on the primary original time domain data (30,000 × 1). Because this study aims to evaluate the selection of signal features for achieving efficient recognition, a TMC model with time domain features as the primary classification features was selected as the comparison model for model evaluation.

The classification results (Tables 5 and 6) reveal that the TMC and TFMC models achieved excellent performance in classifying the blast signal and Gaussian noise signals after suitable training. However, the TFMC model performed better, especially for blast signal recognition; it did not mistakenly identify the latter as seismic signals, and the accuracy for the remaining signals was above 96%.

The ROC curve illustrates the relationship between TPR and FPR. An ideal classifier has an AUC of 1; the closer the AUC is to 1, the better the classifier. The AUC values of the time–frequency model for the MS, blast, and Gaussian noise signals were 0.960, 1, and 0.985, respectively (Figure 6).

**Table 5.** Confusion matrix of the test dataset of TFMC and TMC models. MS, Blast, and Gaussian represent the microfracture, blast, and gaussian noise signals, respectively. The overall accuracies of the TFMC and TMC methods are 98.7% and 96.5%, respectively.

| Model | Prediction Classes | MS | Blast | Gaussian | Accuracy |
|---|---|---|---|---|---|
| TFMC | MS | 486 | 0 | 14 | |
| | Blast | 0 | 500 | 0 | 98.7% |
| | Gaussian | 5 | 0 | 495 | |
| TMC | MS | 473 | 0 | 27 | |
| | Blast | 4 | 496 | 0 | 96.5% |
| | Gaussian | 22 | 0 | 478 | |

**Table 6.** Comparison between the TFMC and TMC methods on the test dataset containing MS, Blast noise, and Gaussian noise data.

| TFMC | | | | | | | |
|---|---|---|---|---|---|---|---|
| Classes | Precision | Recall | Micro F1_Score | Macro F1_Score | TP | FP | FN |
| MS | 0.972 | 0.990 | 0.981 | | 486 | 14 | 5 |
| Blast | 1 | 1 | 1 | 0.987 | 500 | 0 | 0 |
| Gaussian | 0.990 | 0.972 | 0.981 | | 495 | 5 | 14 |
| **TMC** | | | | | | | |
| Classes | Precision | Recall | Micro F1_Score | Macro F1_Score | TP | FP | FN |
| MS | 0.946 | 0.948 | 0.981 | | 473 | 27 | 26 |
| Blast | 0.992 | 0.992 | 0.992 | 0.975 | 496 | 4 | 0 |
| Gaussian | 0.956 | 0.947 | 0.951 | | 478 | 22 | 27 |

Note: Both models are binary classification models (microfracture or noise signal) and, considering the significant difference between the microfracture and blast signals, we artificially classify the microseismic signals classified as noise by the machine-learning model as Gaussian noise for index calculation.

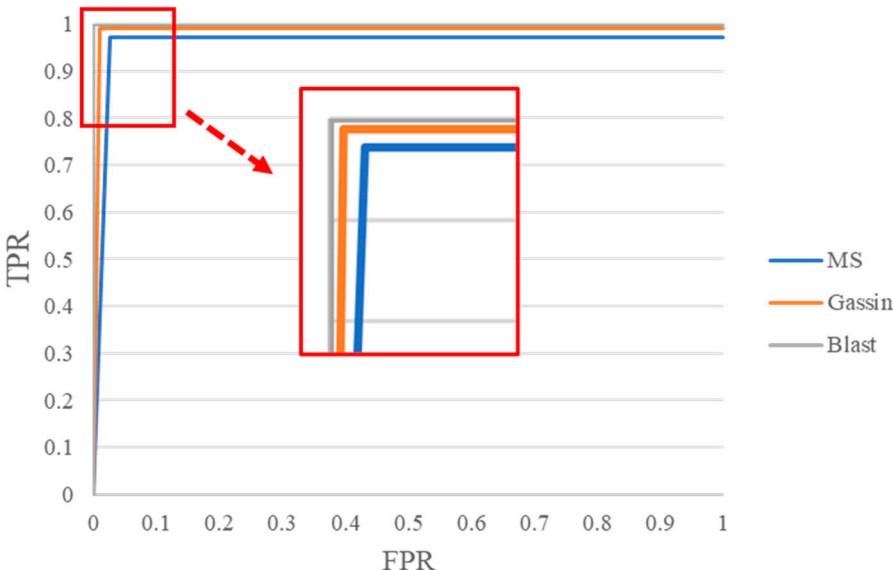

**Figure 6.** The ROC curve of different classes after applying TFMC to the test dataset. The AUC values of the three classes (MS, Gaussian noise, and Blast) are 0.960, 1, and 0.985, respectively.

## 4. Engineering Application and Discussion

### 4.1. Engineering Background

4.1.1. Engineering Geology Overview

The Grand Canyon tunnel of the Lehan Expressway, an ultra-deep buried twin-tube tunnel, is considered for the case study. Its entrance is in Jinkou River District, whereas the exit is in Ganluo County, Liangshan Prefecture. The tunnel is approximately 12.1 km long, and the maximum buried depth reaches 1944 m. Notably, it is the "first buried-deep" highway tunnel in the world. The tunnel mainly passes through a dolomite stratum. The exposure of the palm surface after excavation revealed that the rock stratum occurs almost horizontally, whereas the rock mass structure is a dense horizontally layered structure (the main physical and mechanical parameters of rock mass are listed in Table 7). Due to the considerable buried depth and high ground stress (see Table 8 for the calculation results of the ground stress), drastic stress adjustment occurs in the surrounding rock after excavation disturbance. The sound emanating from inside the rock mass is often audible and accompanied by surface blocks falling off or even ejecting. Due to the influence of complex geological and hydrological factors, rock bursts, water inrush, collapse, and other disasters are highly likely to occur throughout the Grand Canyon tunnel. Field staff often reported frequent weak–moderate rock bursts during its excavation. Only rock bursts that were moderate and above are marked in Figure 7. Notably, some rock bursts occurred more than 40 times from August 2021 to January 2022. Rock burst disasters pose a great threat to personnel and equipment during tunnel construction and affect construction progress. Meanwhile, the energy release caused by high ground stress leads to the rupture and loosening of surrounding rock, thus directly and indirectly affecting the initial support [35] indirectly affecting the second lining, respectively. This in turn poses risks during the operational stage of the tunnel.

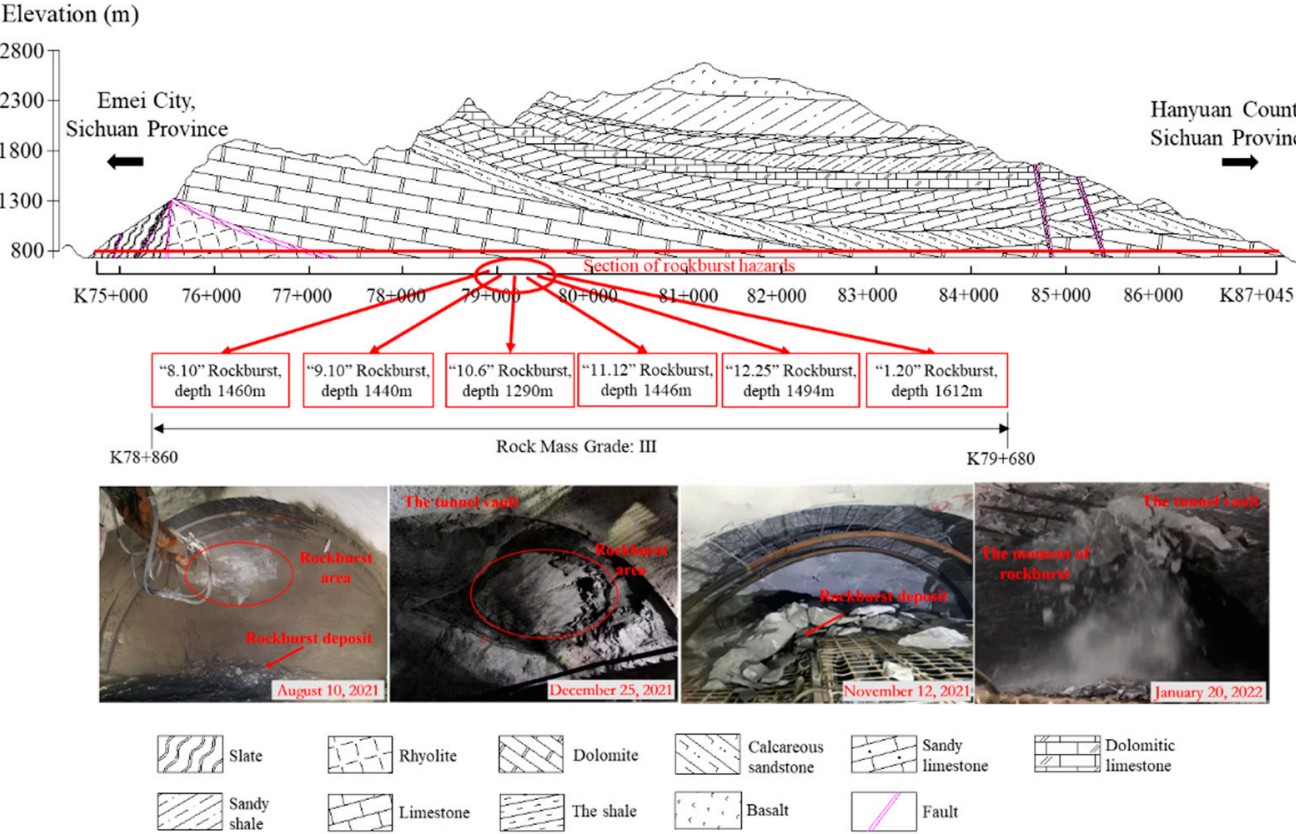

**Figure 7.** Geological cross-sectional view of the Grand Canyon tunnel, and several severe rockburst disasters.

**Table 7.** Physical and mechanical parameters of dolomite.

| Unit | Value |
|---|---|
| Density | 2800–2900 kg/m$^3$, average: 2850 kg/m$^3$ |
| Uniaxial compressive Strength | 46 MPa |
| Poisson ratio | 0.15–0.35 |
| elasticity modulus | $5 \times 10^4$–$9.4 \times 10^4$ MPa |
| Shear modulus | $2.17 \times 10^4$–$3.48 \times 10^4$ MPa, average: $2.82 \times 10^4$ MPa |
| S-wave velocity | 4000 sm/s |

**Table 8.** Calculation results of ground stress.

| Locations | Depth/m | Principal Stress | Value/MPa | Azimuth/° | Dip Angle/° |
|---|---|---|---|---|---|
| K78+923 | 1396.78 | σ1 | 36.61 | 199.76 | −7.21 |
| | | σ2 | 20.64 | 63.84 | −41.39 |
| | | σ3 | 18.49 | 101.76 | −47.70 |

### 4.1.2. Monitoring Sensor Arrays

Rock microfractures are accompanied by elastic waves that spread in the formation medium. Due to the differences in their source location and sensor positions, each sensor receives the signal at different times. Analyzing and calculating the time differences corresponding to at least four sensors allows the source location to be locked in space. Notably, the sensor array must be appropriately designed and installed for accurate positioning. Figure 8 shows the sensor array installed in the field. Due to differences in construction progress, we installed two rows of the sensor array in the advance tube and one in the lag tube; the microseismic sensor arrays arranged on the left and right lines are interoperable. The difference in signal arrival time can be used to quickly determine the location of a microrupture.

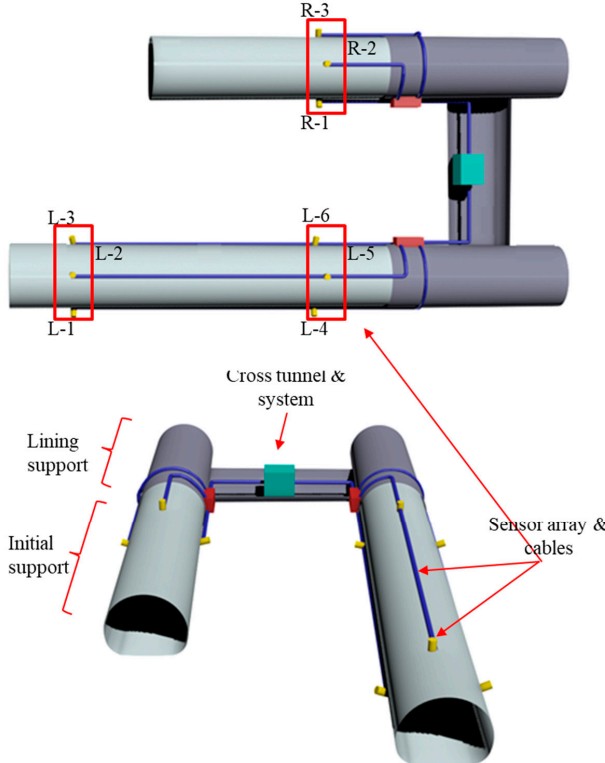

**Figure 8.** Microseismic monitoring system and arrangement of sensor arrays in the twin-tube tunnel.

### 4.2. Intelligent Microseismic Monitoring and Early Warning Based on the Cloud Platform

4.2.1. Microseismic Monitoring Process

As the most effective method of rock burst early warning, microseismic monitoring was utilized during construction of the Grand Canyon tunnel. To realize real-time monitoring throughout the day, our team designed and built a monitoring and warning cloud platform (Figure 9). When an event occurs, the system will automatically generate classification results and source parameters, upload the calculation results to the cloud platform, and generate an early warning report, which the management personnel only need to view on the device terminal. The entire process, from event generation to cloud-platform result presentation, takes approximately 20 s.

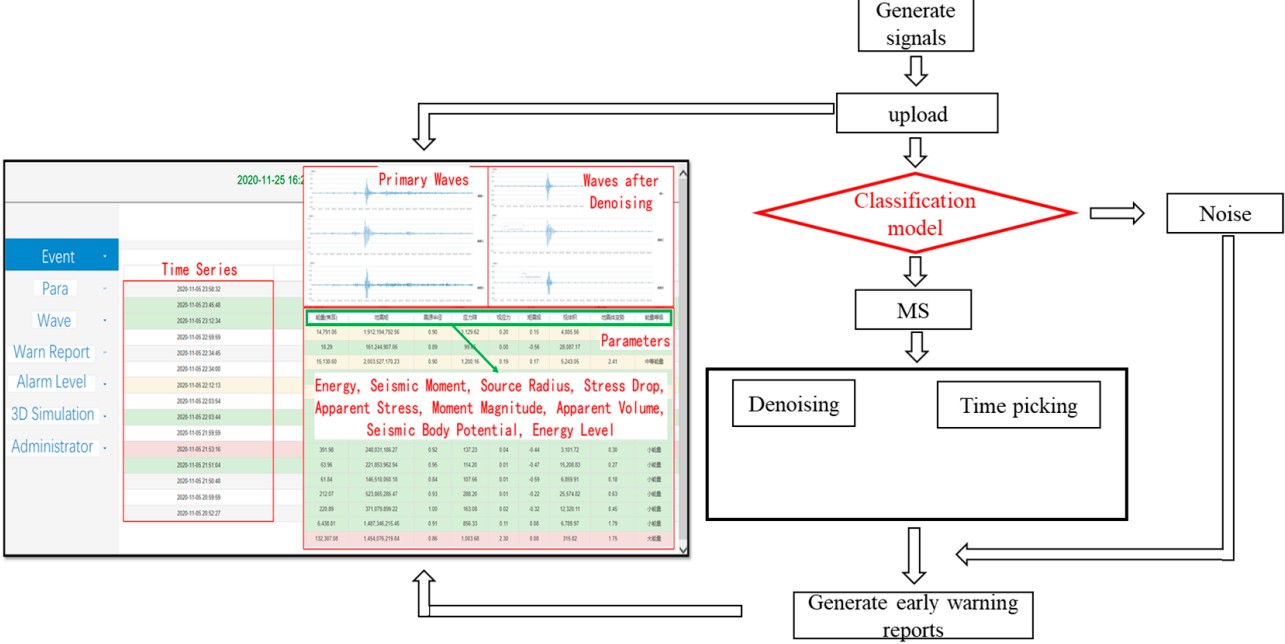

**Figure 9.** Program structure of the early warning system.

4.2.2. Classification Test and Rock Burst Warning

In the microseismic monitoring of the Grand Canyon Tunnel, we identified SN, the cause of which is unclear. Presumably, excavators may produce this noise during tunneling. Similar noise (Figure 10) limits the classification accuracy, which severely impedes routine monitoring and early warning work.

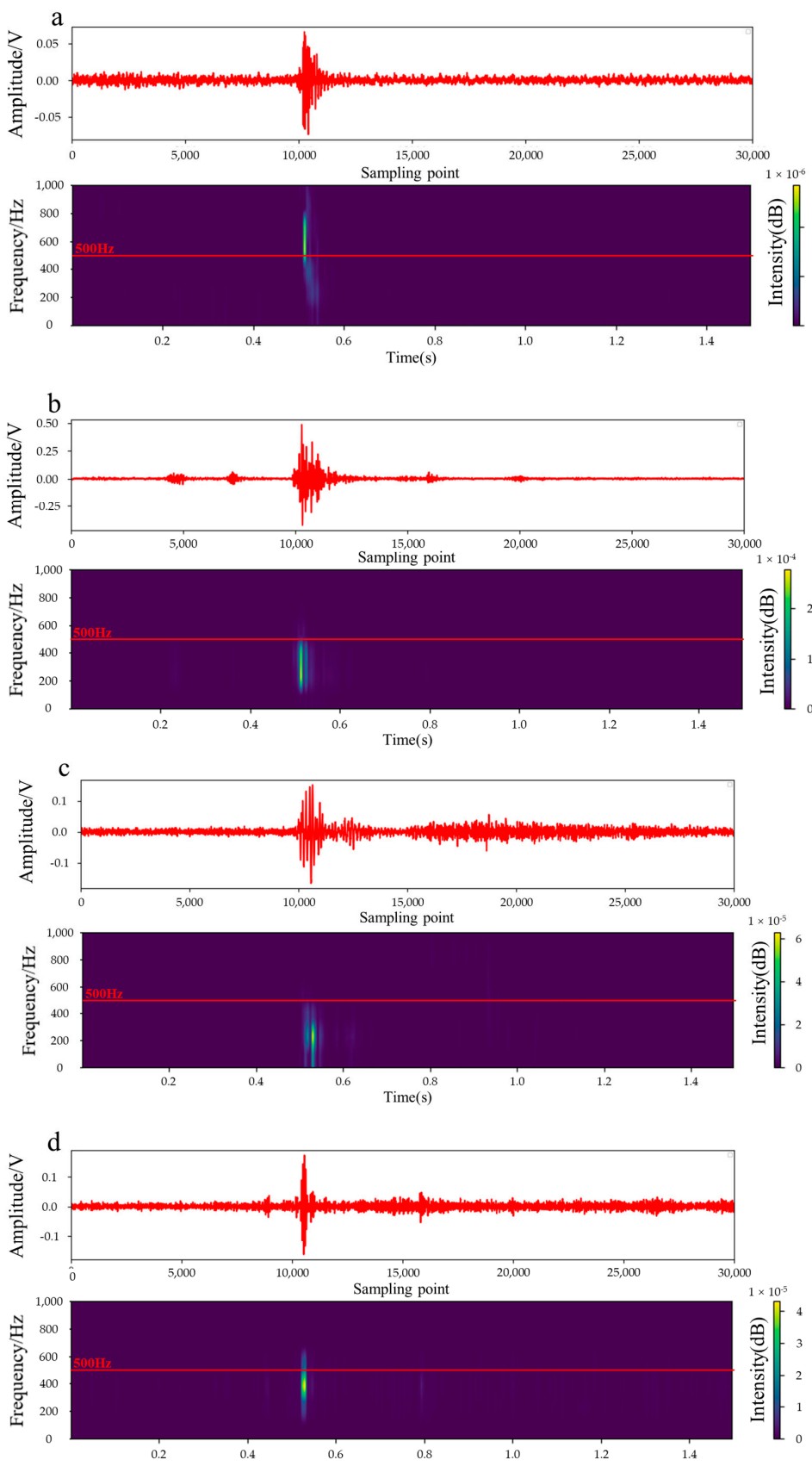

**Figure 10.** Typical low-SNR microseismic signal (**a**), and SN signals (**b**–**d**).

We collected 1000 MS with low SNR and 1000 SN signals to test the classification effect of the time–frequency model. The latter is a binary classification model, and the threshold was set to 0.5 (that is, when the probability the model predicts is higher than 0.5, it is judged as the corresponding category, and when it is less than 0.5, it is judged as the opposite category). The time domain model was retained as the comparison model, and the classification effects of the two models are listed in Tables 9 and 10.

**Table 9.** Comparison of TFMC method and TMC method on the dataset of MS and SN.

| Model | Class | Prediction | |
|---|---|---|---|
| | | MS | SN |
| TFMC | MS | 934 | 66 |
| | SN | 102 | 898 |
| TMC | MS | 958 | 42 |
| | SN | 655 | 345 |

**Table 10.** Comparison between TFMC method and TMC method on the test dataset containing MS and SN.

| TFMC | | | | | | | |
|---|---|---|---|---|---|---|---|
| Classes | Precision | Recall | Micro F1_Score | Macro F1_Score | TP | FP | FN |
| MS | 0.934 | 0.902 | 0.918 | 0.917 | 934 | 66 | 102 |
| SN | 0.898 | 0.932 | 0.915 | | 898 | 102 | 66 |
| TMC | | | | | | | |
| Classes | Precision | Recall | Micro F1_Score | Macro F1_Score | TP | FP | FN |
| MS | 0.958 | 0.594 | 0.733 | 0.615 | 958 | 42 | 655 |
| SN | 0.345 | 0.891 | 0.497 | | 345 | 655 | 42 |

The classification results (Tables 9 and 10) reveal that the TFMC model achieved an accuracy of above 90% in identifying low-SNR MS and SN. In contrast, the TMC model could achieve 95.8% accuracy when identifying low-SNR microseismic signals, but the accuracy was only 34.5% when identifying SN. This is primarily attributable to the differences in the model in learning the relevant features of the training signal. The TMC model only learns the time domain features of the signal, whereas the TFMC model learns not only the time domain features but also the frequency domain features of the signal.

Table 11 presents the event test results, whereas the previous ones were single-signal classification tests. A single event during actual monitoring contains at least three signal waveforms. The standard we made for judging a single event as a microseismic event was "half or more of the signals of the single event need to be identified as microseismic signals". Otherwise, it was regarded as an invalid event, namely, a noise event. On 12 November 2021, a rock burst occurred in the cave, as we predicted. We plan to consider this rock burst event as a case study to evaluate the practical applications of the TMC and TFMC models in disaster warnings in the future. According to the daily monitoring records and manual classification results, the sensor arrays detected 66 microseismic events and 343 noise events 24 h before the rock burst. Table 11 lists the classification results of the TMC and TFMC models. Utilizing the classification results of the latter, we raised an early warning of the rock burst.

**Table 11.** Comparison of TFMC method and TMC method on the classification of events; each event consists of six waveforms.

| Model | Prediction Classes | ME | NE | Acc | Marco-Acc |
|-------|--------|-----|-----|------|-----------|
| TMC | ME | 66 | 0 | 100% | 64.5% |
|  | NE | 145 | 198 | 47% |  |
| TFMC | ME | 62 | 4 | 93.9% | 96.8% |
|  | NE | 9 | 334 | 97.4% |  |

The accuracy of the TMC model in identifying noise events was only 47% (Table 11). Surprisingly, the previously identified SN signals were again observed (Figure 10). Because the TMC model does not learn the frequency domain features of the signal during training, it struggles with effectively distinguishing such SNs. However, the TFMC model successfully learned the frequency domain features after the input data were converted to the time–frequency domain, which makes up for the defects of the time domain model. Accordingly, the overall noise recognition accuracy was improved to 97%. Existing shortcomings are also evident. Although the time–frequency model may miss a few microseismic events, it can generally meet engineering needs. One possible way to solve this omission is to add a misclassification punishment mechanism or compensation, a direction we intend to pursue in a future study.

The excellent performance of the time–frequency model in tunnel microseismic signal recognition was critical to the timely release of early-warning information and the basis of 24 h intelligent monitoring and early warning. Due to the suddenness of rock bursts, the prophase work of microseismic monitoring is critical. The time–frequency domain microseismic classification model exhibited excellent signal classification performance. Moreover, it exhibited good generalization ability and can be applied to signal processing in other fields.

## 5. Conclusions

We propose a microseismic signal classification method based on STFT and deep learning techniques. The model takes the signal after the STFT as the primary training data, after which it is trained by the modified VGG13 network with an attention mechanism. The well-trained classification model can simultaneously extract the time domain and frequency domain features of the signal and has good application prospects in the microseismic monitoring of deep tunnel engineering.

We collected different typical signals, such as microseismic, blasting, and Gaussian environmental noise, and evaluated the classification method through various evaluation indicators. The test results demonstrate that the time–frequency model has better classification performance than the time domain model. Specifically, the recognition accuracy is almost 100% for identifying the blasting signal.

This model is suitable for the fast classification tasks encountered in microseismic real-time monitoring for applications in deep-buried tunnel engineering. Notably, it solves the complex problem of microseismic signal classification in practical engineering. For example, the waveform of a similar noise signal generated by a unique source resembles some microseismic signals with low SNR, and the method can efficiently and accurately identify them.

This microseismic signal classification method based on short-time Fourier time–frequency analysis and deep learning achieves fast calculation and high accuracy. In the microseismic monitoring of deep-buried tunnels, it is beneficial for automatically and intelligently processing massive microseismic data, reducing redundant manual work, and improving the effectiveness of tunnel disaster assessment and early warning. This deep learning-based signal processing method offers high efficiency and can elucidate the

uncertainties that persist in many traditional data-driven fields, such as radar detection signals, seismic waves, and personal identification.

**Author Contributions:** Writing—original draft preparation, C.M.; writing—review and editing, X.R.; methodology, W.X.; formal analysis, W.Y.; supervision, T.L.; data processing, K.D.; data investigation, J.W.; data processing, Y.L.; data investigation, K.T. All authors have read and agreed to the published version of the manuscript.

**Funding:** This research was funded by National Natural Science Foundation of China, grant number 42177173; State Key Laboratory of Geohazard Prevention and Geoenvironment Protection Independent Research Project, grant number SKLGP2020Z010.

**Data Availability Statement:** The dataset for this research is available by email: jamesbond5202@163.com.

**Acknowledgments:** The authors would like to thank all the reviewers who participated in the review, as well as MJEditor (www.mjeditor.com (accessed on 5 December 2022)) for providing English editing services during the preparation of this manuscript.

**Conflicts of Interest:** The authors declare no conflict of interest.

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
