# Peer review of "Fine Classification Method for Massive Microseismic Signals Based on Short-Time Fourier Transform and Deep Learning"

_remotesensing, doi:10.3390/rs15020502_

Round 1

Reviewer 1 Report

The authors proposed a short-time Fourier transform (STFT) and deep learning based method for microseismic signal classification. A comprehensive description has been given, and the manuscript is well organized with high quality figures. It can be accepted after minor revision.

1. How to determine the start and end of the selected signal? Then, is manual picking used? Can automatic method be used?

2. Figure 5: it seems that the validation has a bad performance?

3. The TMC obtains a good accuracy in Table 6, while a bad accuracy for Table 12, Why?

Reviewer 2 Report

The paper is interesting, showing the superior performances of the TFMC approach compared to the TMC approach on real data. I presume that the superior performances are particularly evident because of the particular situation where a non-random noise source is producing signals quite similar to the microseismic target signals in the time domain but well distinguishable in the frequency domain because of a quite different bandwidth. In other situations I had to deal with signal and noise events quite similar also in the frequency domain. That would be a challenging situation also for this approach.

The paper would gain benefit from a language revision by an English mother tongue reviewer.

Specify the meaning of any acronym when it appears in the paper for the first time, e.g., IMF at line 61, ANN at line 79.

Line 33. As you did for the underground applications (mining, powerhouses, tunnel excavations) where you suggest a couple of references, add at least a couple of references for the other projects in which rock mass stability is crucial. I suggest the followings:

Helmstetter, A., & Garambois, S. 2010. Seismic monitoring of Sechilienne rockslide (French Alps): Analysis of seismic signals and their correlation with rainfalls. Journal of Geophysical Research Earth Surface. 115, F03016, https://doi.org/10.1029/2009JF001532.

Zhang, Z., Arosio, D., Hojat, A., Zanzi, L., 2020. Tomographic experiments for defining the 3D velocity model of an unstable rock slope to support microseismic event interpretation, Geosciences. 10, 327, https://doi.org/10.3390/geosciences10090327.

Line 39-40. What is the meaning of unique noise? It is used many times in the paper but never defined. Please specify the meaning here, where it is used for the first time.

Line 43-45. I suggest to add a reference here to a paper that exactly represents an example of the global approach that you are mentioning here:

Zhang, Z., Arosio, D., Hojat, A., Zanzi, L., 2021. Reclassification of Microseismic events through Hypocenter Location: Case Study on an Unstable Rock Face in Northern Italy. Geosciences. 11, 37, https://doi.org/10.3390/geosciences11010037

Line 123 and Table 1. What do you mean by “this experiment” in line 123? And why do you introduce Table 1 with these parameters and then conclude section 2.1 without any comment about these parameters? The details in Table 1 seem not useful, at least at the level of the discussion of section 2.1. I suggest to remove the table and to mention in the discussion only the parameters that really need to be mentioned. Besides, take into account the following comment about these parameters.

I do not understand the meaning of fs and its default value, which is 1. Which frequency is it? The sampling frequency? And what is 1? 1 Hz? I also think that the numbers nperseg, nooverlap and ntft, which are numbers of samples, are useless if the reader does not know what is the frequency range of the target signal and what is the sampling frequency. Finally, there is some confusion about None and False regarding the last parameter.

Figure 1d. The time domain signal is clipped. Can you extend the amplitude range beyond 4V to better represent the signal or this is the max amplitude range recorded by the sensor?  

Line 141-142. I suggest to rephrase as: From the time spectrum analysis, the microseismic signal shows the most energetic components at frequencies higher than 500 Hz.

Line 146-147. I suggest to rephrase as: According to the time spectrum analysis, this type of noise shows the most energetic components at frequencies lower than 500 Hz.

Section 2.3. I wonder why the channel attention mechanism is described and commented by the authors while the spatial attention mechanism is totally ignored. It seems strange to me. Why?

Table 3. Recall and TPR have the same definition. Why two names for the same parameter? Micro-F1 and Macro-F1 requires some explanation about the meaning of the parameter and the meaning of the summation in the Macro-F1 parameter. What is n? It becomes clear later, better to specify here.

Lines 293 and 302 versus Table 6 and 7. Gassin or Gaussian? Is it a typo? Gassin as a shortcut of Gaussian?

Figure 6. It suddenly appears at the end of section 3.4 without any comment in the discussion of the result. Remove or use it to enrich the discussion.

Lines 349-350. I disagree. Three sensors are not enough to locate the hypocenter. They are enough if you know the velocity and the absolute time of the event. If you measure the time differences, you need 4 or 5 sensors depending on your assumptions about the velocity.

Line 396. Table 7 is actually Table 12.

List of References. Reference [30] is not referred to in the manuscript.
